# Single Local Injection of Epigallocatechin Gallate-Modified Gelatin Attenuates Bone Resorption and Orthodontic Tooth Movement in Mice

**DOI:** 10.3390/polym10121384

**Published:** 2018-12-13

**Authors:** Yuta Katsumata, Hiroyuki Kanzaki, Yoshitomo Honda, Tomonari Tanaka, Yuuki Yamaguchi, Kanako Itohiya, Sari Fukaya, Yutaka Miyamoto, Tsuyoshi Narimiya, Satoshi Wada, Yoshiki Nakamura

**Affiliations:** 1Department of Orthodontics, School of Dental Medicine, Tsurumi University, 2-1-3 Tsurumi, Tsurumi-ku, Yokohama, Kanagawa 230-8501, Japan; yutakatsumata0904@gmail.com (Y.K.); yamaguchiyuki0911@gmail.com (Y.Y.); kasuya-kanako@tsurumi-u.ac.jp (K.I.); fukaya-sari@tsurumi-u.ac.jp (S.F.); miyamoto-y@tsurumi-u.ac.jp (Y.M.); narimiya-tsuyoshi@tsurumi-u.ac.jp (T.N.); wada-s@tsurumi-u.ac.jp (S.W.); nakamura-ys@tsurumi-u.ac.jp (Y.N.); 2Institute of Dental Research, Osaka Dental University, 8-1 Kuzuhahanazonocho, Hirakata, Osaka 573-1121, Japan; honda-y@cc.osaka-dent.ac.jp; 3Department of Biobased Materials Science, Graduate School of Science and Technology, Kyoto Institute of Technology, Matsugasaki, Sakyo-ku, Kyoto 606-8585, Japan; t-tanaka@kit.ac.jp

**Keywords:** osteoclast, RANKL, ROS, antioxidant enzyme, Nrf2, gelatin, epigallocatechin gallate

## Abstract

Osteoclastic bone resorption enables orthodontic tooth movement (OTM) in orthodontic treatment. Previously, we demonstrated that local epigallocatechin gallate (EGCG) injection successfully slowed the rate of OTM; however, repeat injections were required. In the present study, we produced a liquid form of EGCG-modified gelatin (EGCG-GL) and examined the properties of EGCG-GL with respect to prolonging EGCG release, NF-E2-related factor 2 (Nrf2) activation, osteoclastogenesis inhibition, bone destruction, and OTM. We found EGCG-GL both prolonged the release of EGCG and induced the expression of antioxidant enzyme genes, such as heme oxygenase 1 (Hmox1) and glutamate-cysteine ligase (Gclc), in the mouse macrophage cell line, RAW264.7. EGCG-GL attenuated intracellular reactive oxygen species (ROS) levels were induced by the receptor activator of nuclear factor-kB ligand (RANKL) and inhibited RANKL-mediated osteoclastogenesis in vitro. An animal model of bone destruction, induced by repeat Lipopolysaccharide (LPS)-injections into the calvaria of male BALB/c mice, revealed that a single injection of EGCG-GL on day-1 could successfully inhibit LPS-mediated bone destruction. Additionally, experimental OTM of maxillary first molars in male mice was attenuated by a single EGCG-GL injection on day-1. In conclusion, EGCG-GL prolongs the release of EGCG and inhibits osteoclastogenesis via the attenuation of intracellular ROS signaling through the increased expression of antioxidant enzymes. These results indicate EGCG-GL would be a beneficial therapeutic approach both in destructive bone disease and in controlling alveolar bone metabolism.

## 1. Introduction

Osteoclasts are multi-nucleated cells that can resorb bone tissue [1]. They are tightly regulated by the receptor activator of nuclear factor-kB ligand (RANKL) [2]. Excessive activation of osteoclasts is the primary cause of destructive bone diseases, such as periodontitis [3]. In addition to pathological bone destruction, osteoclasts are also involved in physiological bone remodeling [4]. Osteoclastic bone resorption in the compression zone of the periodontal ligament (PDL) enables the movement of teeth during orthodontic treatment [5,6,7,8,9,10].

The regulation of orthodontic tooth movement (OTM) by RANKL has been extensively investigated. RANKL expression is upregulated in the compression zone of the PDL during OTM [11,12] and is increased in the gingival crevicular fluid in patients undergoing orthodontic treatment [13,14]. In addition, induction of local RANKL by gene transfer [15] and vibration [16] accelerates OTM in animal models. Osteoprotegerin (OPG), a decoy receptor for RANKL, can negatively regulate OTM where local delivery of OPG [17] and localized OPG-gene induction by gene transfer [18] inhibits OTM. Although this local gene transfer is an effective way to control the rate of OTM, it is difficult to apply clinically. Therefore, safe pharmacological regulation of the rate of OTM is needed.

A reactive oxygen species (ROS) is an intracellular signaling molecule downstream of RANKL [19]. Consequently, scavenging ROS is a promising approach for osteoclast inhibition. Activation of nuclear factor E2-related factor 2 (Nrf2) induces antioxidant enzymes that inhibit osteoclastogenesis [20,21,22,23,24]. EGCG can induce Nrf2-mediated anti-oxidation and ROS scavenging, and consequently, inhibit the rate of OTM [21]. However, repeat injections of EGCG are necessary to successfully inhibit the rate of OTM via the suppression of osteoclastogenesis.

We previously fabricated an EGCG-modified gelatin sponge (EGCG-GS), where a single EGCG-GS implantation could successfully induce osteogenesis at the artificial bone defect in mice calvaria [25]. Therefore, we hypothesized that a single injection of EGCG-GS in liquid form (EGCG-GL) would be able to maintain an adequate local concentration of EGCG long enough to inhibit osteoclastogenesis in experimental animals. In this study, we examined the ability of prolonged EGCG release from EGCG-GL and confirmed that a single injection of EGCG-GL could inhibit osteoclastogenesis in vivo.

## 2. Materials and Methods

### 2.1. Chemicals

Type A gelatin from porcine skin was purchased from Sigma-Aldrich Co. LLC. (St. Louis, MO, USA). EGCG was purchased from BioVerde Inc. (Kyoto, Japan). The reagents, 4-(4,6-dimethoxy-1,3,5-triazin-2-yl)-4-morpholinium chloride (DMT-MM) and N-methylmorpholine (NMM), were purchased from Tokyo Chemical Industry Co., Ltd. (Tokyo, Japan) and Nacalai Tesque Inc. (Kyoto, Japan), respectively.

### 2.2. Preparation of EGCG-GL

A detailed protocol for preparing EGCG-GS has been previously described [25]. EGCG-GL was prepared using the protocol for preparing EGCG-GS with a slight modification. Briefly, gelatin (100 mg) was dissolved in warm water (5 mL) at 50 °C. After the solution was cooled to room temperature, NMM (27.5 µL), EGCG (0.07 or 0.7 mg), and DMT-MM (69.2 mg) were added, and the solution was stirred for 24 h at room temperature in the dark. The products were purified by dialysis (Spectra/Por7 MWCO 1000, Spectrum Labs, Rancho Dominguez, CA, USA) in water in the dark. Solutions of low EGCG-GL (0.07 mg EGCG/10 mL solution) and high EGCG-GL (0.7 mg EGCG/10 mL solution) were prepared. These prepared EGCG-GL remained liquid though they had a slight viscosity.

### 2.3. Examination of Prolonged Release of EGCG from EGCG-GL

Using high EGCG-GL (0.7 mg EGCG/10 mL solution) and high EGCG solution (0.7 mg EGCG/10 mL solution), bromelain (100 µg) was added into 500 µL of each test solution and incubated at 37 °C for 4 h. The solutions were then separated using a 50 kDa molecular weight cut-off centrifugal filter (EMD Millipore, Billerica, MA, USA), and the flow-through was collected to measure the released EGCG. The filter was then resuspended in 500 µL PBS, incubated with another 100 µg bromelain, and separated three more times for a total of four flow-throughs for each sample. The average molecular weight of gelatin is from 50 to 100 kDa, and that of EGCG is 458 Da. Therefore, our filtration can separate released EGCG from the EGCG-gelatin complex.

The concentration of EGCG in the flow-through was measured with direct Enzyme-Linked ImmunoSorbent Assay (ELISA) using a mouse monoclonal anti-catechin antibody (Cosmo Bio Inc., Tokyo, Japan). Briefly, samples and standards were immobilized in wells for 2 h then washed with PBS. The wells were then blocked with BlockAce (DS Pharma Biomedical Co., Ltd., Osaka, Japan) for 2 h, probed with diluted mouse IgG anti-catechin antibody and incubated for 2 h, washed, then probed with an horseradish peroxidase (HRP)-conjugated anti-mouse IgG secondary antibody for 2 h. Color development was performed using 3,3′,5,5′-tetramethylbenzidine (TMB) solution, and optical density measured by a plate reader (BioTek, Tokyo, Japan). EGCG standards were serial two-fold dilutions from 70 µg/mL, and the sensitivity (defined as the concentration of analyte giving an absorbance higher than the mean absorbance of the blank plus three standard deviations) was 0.79 µg/mL. The coefficient of variance value of the assay at 70 µg/mL and 2.2 µg/mL was 5.2% and 23.5%, respectively.

### 2.4. Cells and Culturing

The mouse monocytic cell line, RAW 264.7, was obtained from the Riken Bioresource Center (Tsukuba, Japan). The cells were cultured in alpha modified Eagle’s medium (Wako Pure Chemical, Osaka, Japan) with 10% fetal bovine serum (FBS; Thermo Fisher Scientific, Waltham, MA, USA) supplemented with penicillin (100 U/mL) and streptomycin (100 µg/mL). All cells were cultured at 37 °C in a 5% CO_2_ incubator.

### 2.5. Cytotoxicity Assay of EGCG-GL

Cytotoxicity was assessed using an AlamarBlue Cell Viability reagent (ThermoFisher, Tokyo, Japan). In brief, RAW 264.7 cells were plated in 24-well plates and cultured with the various dilutions of EGCG-GL for 1 day. We used EGCG-GL at dilutions of 1/100, 1/1000, 1/10000, 1/100000, and no EGCG-GL, which gave final EGCG concentrations of 0.7, 0.07, 0.007, 0.0007, and 0 mg/L, respectively. The kit reagent was then added to the cultures, and fluorescence (excitation: 545 nm, emission: 590 nm) was measured using the Synergy HTX Multi-Mode plate Reader (BioTek Japan, Tokyo, Japan) after 2 h incubation.

### 2.6. Intracellular ROS Detection

RAW 264.7 cells were pretreated with or without EGCG-GL (0.07 mg/L) for one day then stimulated with recombinant RANKL (100 ng/mL) for 6 h, washed, and harvested. The cell suspension was then incubated on ice for 30 min with a fluorescent superoxide probe (BES-So-AM, 5 µM final; Wako Pure Chemical) in PBS containing 2% FBS. After three washes with PBS containing 2% FBS, intracellular ROS was detected using an AccuriC6 flow cytometer (BD Biosciences, San Jose, CA, USA), and data were processed using FlowJo analysis software (FlowJo, LLC, Ashland, OR, USA). The viable cellular fraction of the monocyte/macrophage was gated on a forward scatter/side scatter plot, and intracellular ROS levels were monitored in the FL-1 channel.

### 2.7. Real-Time RT-PCR Analysis

For antioxidant gene expression analysis, RNA was extracted two days after the treatment of cells with EGCG-GL or gelatin solution, and the expression of Nrf2, heme oxygenase 1 (Hmox1), and glutamate-cysteine ligase (Gclc) were analyzed. RNA was extracted from RAW 264.7 cells using the GenElute mammalian total RNA Miniprep-kit (Sigma-Aldrich, St. Louis, MO, USA) with an on-column genomic DNA digestion. Isolated RNA (500 ng each) was reverse-transcribed with an iScript cDNA-Supermix (Bio-Rad, Hercules, CA, USA). Real-time RT-PCR was performed with SsoFast™ EvaGreen^®^ Supermix EvaGreen-Supermix (Bio-Rad). PCR primers were from PrimerBank (Boston, MA, USA) and have been described previously [21,23,24]. Fold changes of gene expression were calculated by using the −∆∆Ct method with ribosomal protein S18 (RPS18) as the reference gene.

### 2.8. Osteoclastogenesis Assay

RAW 264.7 cells were plated on 24-well plates (5 × 10^3^ cells/well) in the presence or absence of recombinant RANKL (100 ng/mL) and EGCG-GL. After four days of culture, cells were stained for Tartrate-resistant acid phosphatase (TRAP) using an acid phosphatase kit (Sigma-Aldrich). Dark red multi-nucleated cells (over three nuclei) were counted as TRAP-positive, multi-nucleated cells.

### 2.9. Experimental Animals

All animal experimental protocols were reviewed and approved by the Institutional Animal Care and Use Committee of Tsurumi University (28A031 for the OTM experiment and 28A087 for the bone destruction experiment). Animal experiments were performed in compliance with the Regulations for Animal Experiments and Related Activities at Tsurumi University. Seven, week-old male BALB/C mice (Clea Japan, Tokyo, Japan) were used for both experiments.

### 2.10. Bone Destruction Model

Repeat injections of Lipopolysaccharide (LPS) were used for the in vivo bone destruction model [20,23,26]. Mice were randomly assigned to four equal groups (n = 5 each); a PBS-injected group (control group), an EGCG-GL-injected group (EGCG-GL group), an LPS-induced bone resorption group (LPS group), and an LPS-induced bone resorption and EGCG-GL-injected group (LPS + EGCG-GL group). Purified LPS from Escherichia coli O111:B4 (Sigma-Aldrich) was dissolved in PBS at a concentration of 1 mg/mL. On day 1, 10 µL of LPS solution with 2 µL of gelatin solution or EGCG-GL (140 ng EGCG) were injected into the animals of the LPS and LPS + EGCG-GL groups, respectively. Similarly, 10 µL of PBS with 2 µL of gelatin solution or EGCG-GL (140 ng EGCG) were injected into the animals in the control and EGCG-GL groups on day 1, respectively. Ten microliters of LPS solution or PBS were injected on day 3, 5, 7, and 9. Injections were performed under anesthesia with a 30-gauge needle at a point on the midline of the skull located between the eyes. On day 11, mice were euthanized by cervical dislocation, and cranial tissue samples were fixed overnight with 4% paraformaldehyde in PBS.

### 2.11. Micro-Computed Tomography (microCT) Analysis for Bone Destruction

Fixed cranial tissue samples were subsequently scanned with an X-ray microCT system (inspeXio SMX-225CT; Simadzu Corp., Kyoto, Japan). The scanning parameters were as follows; voxel size: 33 micrometer, slice thickness: 58 micrometer, tube voltage: 70 kV, tube current: 160 mA. After reconstitution, the DICOM files were rendered into three-dimensional images using the Pluto software (http://pluto.newves.org/trac). The percentage of resorbed area was calculated from the ratio of the number of pixels in the resorbed area in the cranial bone to the number of pixels in the total analyzed image of the cranial bone, using ImageJ software (National Institutes of Health, Bethesda, MD, USA). The region of interest was set between the fronto-parietal (coronal) suture and parieto-occipital (lamboidal) suture.

### 2.12. OTM

The OTM-experiment used a total of 13 mice divided into three groups. Group-1 consisted of four mice injected with gelatin solution (control group). Groups-2 and -3 consisted of four and five mice, respectively, injected with an EGCG-GL. Group-2 (low EGCG-GL group) used 0.07 mg EGCG/10 mL solution and Group-3 (high EGCG-GL group) used 0.7 mg EGCG/10 mL solution. In each group, 5 µL of the solution was injected into the sub-periosteal area, adjacent to the maxillary right-side first-molars, on day-1 of the experiment.

The method used for the application of orthodontic-force was slightly modified from a previously described protocol [21]. Briefly, a uniform, standardized compressive-spring, made of 0.012-inch nickel-titanium wire (Nitinol Classic, 3 M Unitek, Monrovia, CA, USA), was placed between the right and left maxillary first-molars, causing them to move palatally. The appliances were introduced into the experimental animals under anesthesia (intraperitoneal-injection of the mixture of medetomidine (0.3 mg/kg), midazolam (4 mg/kg), and butorphanol-tartrate (5 mg/kg)). The movement of the molars was measured every week during the experiment, as described previously [18], and the amount of OTM between groups was compared (method error: 0.011 mm). Animals were euthanized with cervical dislocation on day-21, and the upper-jaw (including molars) was dissected, with the specimen fixed with 4%-paraformaldehyde in PBS overnight.

### 2.13. Immunohistological and Histological Examinations

The bone destruction experiment specimen, calvarial bones, were decalcified with 10% ethylenediamine-tetraacetic acid in PBS for 3 weeks at 4 °C, dehydrated, and embedded in paraffin. Serial frontal sections (6 µm thick) were then prepared and used for the detection of the oxidative stress marker. Briefly, the sections were deparaffinized, treated with LAB solution (Polysciences Inc., Warrington, PA, USA) for antigen liberation, had their endogenous peroxidase activity blocked (3% hydrogen peroxide), were preincubated in 4% BlockAce (DS Pharma Biomedical Co., Ltd., Osaka, Japan), and were subsequently incubated with anti-8-OHdG polyclonal antibody (Bioss Inc., Woburn, MA, USA). After being rinsed, the sections were incubated with the peroxidase-conjugated secondary antibody (ThermoFisher Scientific Inc., Waltham, MA, USA). To visualize the immunoreactivity, the sections were flooded with a diaminobenzidine solution (Vector Laboratories, Burlingame, CA, USA).

OTM experimental specimens were decalcified, dehydrated, and embedded in paraffin. Periodontal tissues from the mesial root of the upper first molars were examined in serial cross sections (6 µm thick) of the molars at the depth from bifurcation level to root apex. TRAP-staining of the sections was performed for osteoclast detection and TRAP-positive cells that formed resorption lacunae on the compression-side of the alveolar bone surface adjacent to the mesial root of the upper first molars were counted.

### 2.14. Statistical Analysis

All data are presented as the mean ± standard deviation (SD). Multiple comparisons (a priori comparisons) were performed using Tukey’s test. *p* < 0.05 was considered to be statistically significant.

## 3. Results

### 3.1. EGCG-GL Prolongs EGCG Release

We first examined whether EGCG-GL can maintain a prolonged release of EGCG, see Figure 1. EGCG solution gave a higher concentration of EGCG compared to that of EGCG-GL in the first flow-through fraction. However, while the EGCG concentration reduced to almost zero in the subsequent flow-through fractions of EGCG solution, EGCG was detected in the subsequent fractions even in the fourth flow-through of EGCG-GL. These results indicate that EGCG-GL has the capacity to prolong the release of EGCG.

### 3.2. EGCG-GL Exhibited No Cytotoxicity against RAW 264.7 Cells at the Concentrations We Tested

We then examined whether EGCG-GL exhibits cytotoxicity against RAW 264.7 cells (Figure 2). We tested various concentrations of EGCG-GL up to 0.7 mg/L and found there was no cytotoxicity against the cells. Therefore, we decided to use EGCG-GL at 0.07 mg/L for in vitro assay hereafter.

### 3.3. EGCG-GL Induces Anti-Oxidant Gene Expression in RAW 264.7 Cells

We then examined the expression of antioxidant genes in RAW 264.7 cells following treatment with EGCG-GL, see Figure 3. We observed no microscopic difference between the control cells and the cells treated with EGCG-GL (data not shown). Compared with the control (addition of gelatin solution), EGCG-GL induced the expression of Nrf2, Gclc, and Hmox1, see Figure 3a–c, respectively. There was no difference between the untreated control and gelatin treatment (data not shown). These results suggest that EGCG-GL augments Nrf2-mediated antioxidant gene expression.

### 3.4. EGCG-GL Attenuates RANKL-Mediated Intracellular ROS

Next, we investigated whether EGCG-GL could interfere with RANKL-mediated intracellular ROS production in RAW 264.7 cells, see Figure 4, using flow cytometry. The viable cellular fraction of monocytes/macrophages was gated on a forward scatter/side scatter plot, see Figure 4a, and intracellular ROS levels were monitored in the FL-1 channel, see Figure 4b. Treatment of RAW 264.7 cells with RANKL (100 ng/mL) increased the intracellular production of the superoxide, as detected using BES-So-AM, and treatment with EGCG-GL inhibited the RANKL-mediated increase of intracellular ROS, see Figure 4b. There was no difference between RANKL treatment and RANKL + gelatin treatment (data not shown). These results suggest EGCG-GL attenuates RANKL signaling via intracellular ROS production.

### 3.5. EGCG-GL Inhibits RANKL-Mediated Osteoclastogenesis

RANKL stimulation induced the formation of TRAP-positive multinucleated cells from RAW 264.7 cells, see Figure 5c,g. The addition of EGCG-GL to the culture significantly inhibited RANKL-mediated osteoclastogenesis, see Figure 5d,h, with a statistically significant difference in the number of TRAP + multi-nucleated cells between RANKL and RANKL + EGCG-GL treatments, see Figure 5i. These results suggest EGCG-GL inhibits RANKL-mediated osteoclastogenesis by attenuating intracellular ROS signaling after RANKL stimulation.

### 3.6. Local Single EGCG-GL Injection Inhibited LPS-Mediated Oxidative Stress in Mice

As Hsu et al. reported that LPS induces oxidative stress [27], we examined whether local single EGCG-GL injection inhibited LPS-mediated oxidative stress or not. Immunohistological staining for 8-OHdG, the marker for oxidative stress, confirmed that local LPS injection induced oxidative stress in mice, see Figure 6a,b. A local, single EGCG-GL injection attenuated LPS-mediated oxidative stress, see Figure 6b,c. There was no statistically significant difference between the control and the LPS + EGCG-GL group, as shown in Figure 6d. These data suggested that the local single EGCG-GL injection blocked oxidative stress in situ.

### 3.7. Local Single EGCG-GL Injection Ameliorates LPS-Induced Bone Destruction in Mice

We then examined whether a single EGCG-GL injection could ameliorate bone destruction in mice calvaria. Repeat LPS injections (five times, every other day, for a total of 50 µg of LPS) induced bone destruction in mice, compared with the control group, see Figure 7a,c. A single local EGCG-GL injection ameliorated LPS-mediated bone destruction as demonstrated by microCT imaging of resorbed areas in calvaria, see Figure 7c–e. These results indicate that a local single EGCG-GL injection inhibits osteoclasts in vivo and that EGCG-GL is a potential inhibitor of bone destruction.

### 3.8. Local Single EGCG-GL Injection Inhibits the Rate of OTM via Attenuation of Osteoclastogenesis

Finally, we examined the effect of a single injection of EGCG-GL on tooth movement in mice, see Figure 8a. The gelatin solution had no impact on OTM, which is consistent with our previous findings [21]. A single injection of EGCG-GL inhibited the rate of OTM for the entire experimental period. A high concentration of EGCG in EGCG-GL exhibited stronger inhibition compared with a low concentration of EGCG in EGCG-GL. These results suggest that a single injection of EGCG-GL inhibits the rate of OTM in a dose-dependent manner.

TRAP staining revealed that OTM causes a certain level of osteoclastogenesis on the alveolar bone surface in the compression zone of PDL, see Figure 8b. Gelatin had no impact on osteoclastogenesis, with many osteoclasts observed on the alveolar bone surface, as shown in Figure 8b. However, a single injection of EGCG-GL significantly reduced the number of osteoclasts on the alveolar bone surface relative to controls, see Figure 8c,d. These results suggest that a single injection of EGCG-GL inhibits osteoclastogenesis, and thereby attenuates OTM.

## 4. Discussion

We previously reported that repeated local injections of EGCG solution inhibited osteoclastogenesis, and consequently, retards OTM [21]. However, repeated local injections are not a realistic approach to orthodontic treatment. In this study, we addressed this issue by developing EGCG-GL.

A single injection of EGCG-GL inhibited osteoclastogenesis in vivo and thereby attenuated OTM in mice, similar to that of the repeated injection of EGCG solution [21]. EGCG-GL would be beneficial for the orthodontic patients to enforce anchorage value by reducing the rate of OTM. EGCG-GL prolonged the release of EGCG and maintained an effective dose of EGCG in the compression zone of PDL during the experimental period. These observations were supported by in vitro results showing an effective dose of EGCG was maintained by EGCG-GL with an augmentation in the expression of antioxidant enzymes.

EGCG-GL inhibited osteoclastogenesis in vitro via attenuation of intracellular ROS signaling by increasing the expression level of antioxidant enzymes. Two major antioxidant enzymes, Hmox1 [28] and Gclc [29], are transcriptionally regulated by Nrf2 [21], and increased by EGCG-GL. We previously reported that EGCG augments the expression of antioxidant enzymes via transcriptional activation of Nrf2 [21]. The results in the current study agree with an earlier report that the induction of antioxidant enzymes via EGCG activation of Nrf2 was an important molecular target for cancer chemoprevention [30].

Although we discovered a direct inhibitory effect of EGCG upon osteoclasts, a possible indirect inhibitory effect of EGCG on osteoclasts should also be considered. EGCG suppresses LPS-induced expression of RANKL in osteoblasts [31]. In addition, EGCG synergistically potentiates prostaglandin-stimulated synthesis of OPG in osteoblasts [32,33]. As a result, EGCG decreases the RANKL/OPG ratio at the site, which indirectly inhibits osteoclastic differentiation. Further investigations are necessary to clarify whether EGCG indirectly inhibits osteoclastic differentiation in vivo.

EGCG-GL was generated by chemically cross-linking EGCG and gelatin through a simple and environmentally friendly synthetic method [25], while maintaining the activity of EGCG [34]. Bromelain, a mixture of proteolytic enzymes derived from pineapples, is added to EGCG-GL to maintain the prolonged release of EGCG by the gradual degradation of gelatin. Physiologically, there are several enzymes that degrade gelatin in the body, such as matrix metalloproteinases (MMPs) [35,36]. Interestingly, MMPs are increased by inflammation [36] and OTM [37], which suggest the release of EGCG from EGCG-GL would be increased in inflammatory sites such as a bone destructive condition and OTM.

Chitosan and collagen, chemically cross-linked with EGCG, were used in another study [38] and it was found that blending gelatin with chitosan improved biodegradability, compared to pure chitosan [39]. Other possible methods for the prolonged release of EGCG are the encapsulation of EGCG into nanoparticles [40], loading EGCG onto poly lactic-co-glycolic acid [41] and double-layered hydroxide nanoparticles [42]. In these methods, EGCG was not chemically cross-linked with the base material, as this is not considered to prolong EGCG release. Our previous study demonstrated that gelatin mixed but not cross-linked with EGCG failed to prolong the release of EGCG, compared with chemically cross-linking gelatin with EGCG [25].

We measured the concentration of released EGCG in each fraction by the ELISA method, to explain, in detail, the kinetic mode, which can measure high concentration and middle to low concentration in one plate. However, the kinetic mode made the calculation of the amount of released EGCG difficult. We could not calculate the extent of EGCG release in the latter fractions, but our data clearly demonstrated the prolonged release of EGCG from EGCG-GL.

In conclusion, we have discovered that EGCG-GL maintains a prolonged release of EGCG and inhibits osteoclastogenesis via attenuation of intracellular ROS signaling by augmenting antioxidant enzymes. EGCG-GL would be a novel therapeutic method to prevent bone destructive disease and, additionally, to modulate tooth movement in orthodontic treatment.

## Figures and Tables

**Figure 1 polymers-10-01384-f001:**
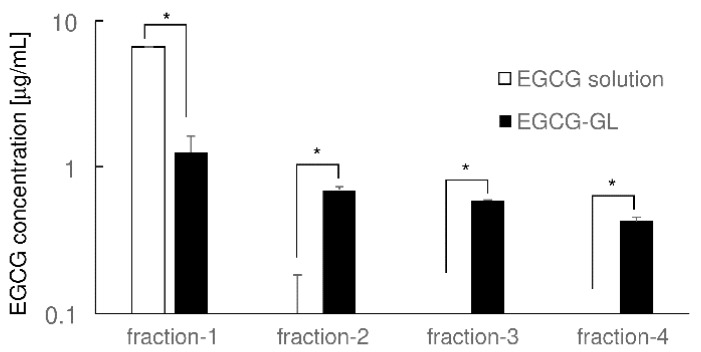
EGCG-GL prolonged the release of EGCG. EGCG concentration in each sample was measured by ELISA (n = 3). *: *p* < 0.05 between samples. Epigallocatechin gallate (EGCG); EGCG-modified gelatin (EGCG-GL).

**Figure 2 polymers-10-01384-f002:**
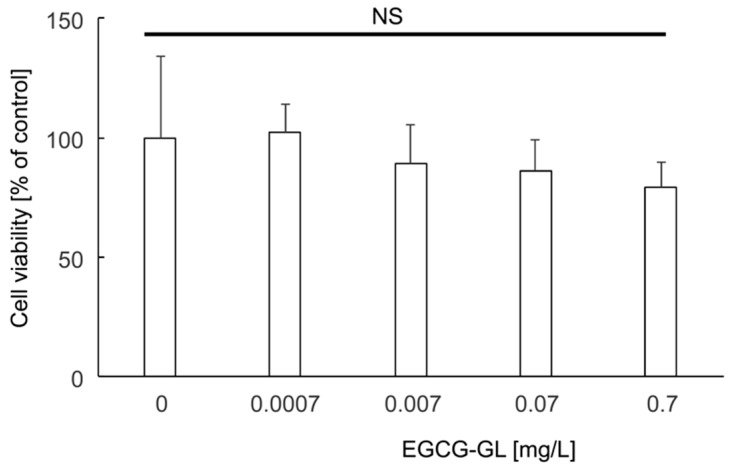
EGCG-GL exhibited no cytotoxicity against RAW 264.7 cells at the concentrations we tested. Percent of cell viability to the control is shown (n = 3). NS: No significant difference among the groups.

**Figure 3 polymers-10-01384-f003:**
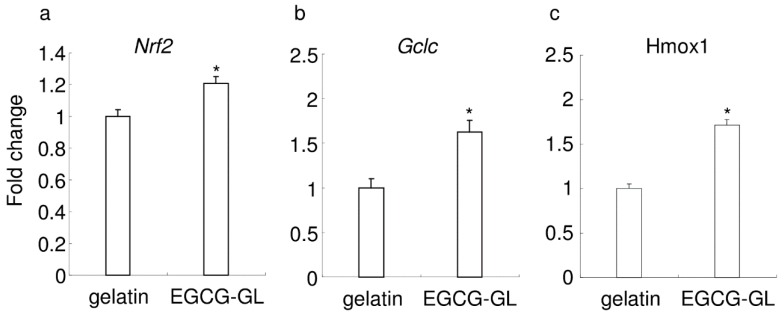
EGCG-GL augmented the expression of antioxidant enzyme genes. Gene expression for Nrf2 (**a**), glutamate-cysteine ligase (Gclc) (**b**), and heme oxygenase 1 (Hmox1) (**c**). (n = 3) *: *p* < 0.05 versus control (sample treated with gelatin solution).

**Figure 4 polymers-10-01384-f004:**
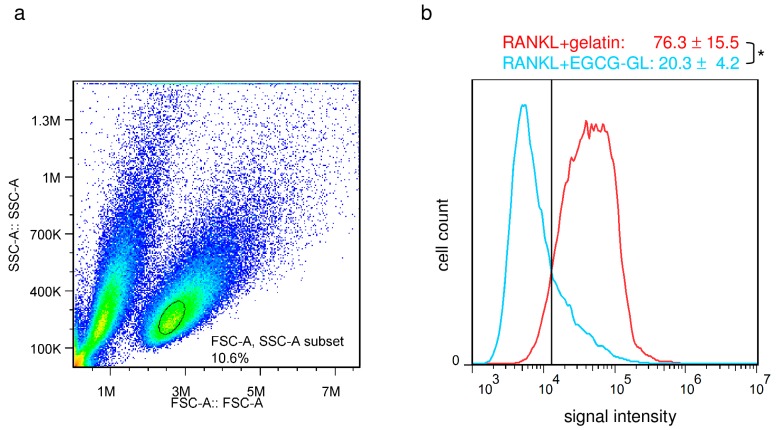
EGCG-GL attenuates the receptor activator of nuclear factor-kB ligand (RANKL)-mediated intracellular reactive oxygen species (ROS) levels. (**a**) The viable cellular fraction of monocytes/macrophages was gated on a forward scatter/side scatter plot (black circle). (**b**) Intracellular ROS levels in RANKL-treated (red) and RANKL- and EGCG-GL-treated RAW 264.7 cells (blue). The vertical line indicates a conventional threshold for ROS-negative and -positive populations. Mean percent of ROS-positive cells are shown on top. (n = 3) *: *p* < 0.05 between samples.

**Figure 5 polymers-10-01384-f005:**
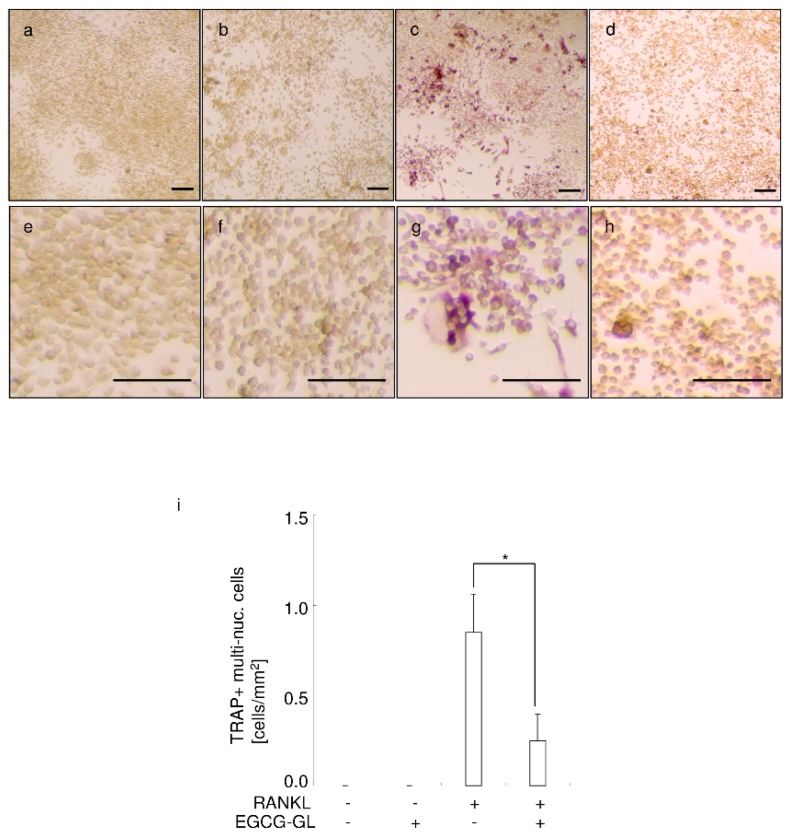
EGCG-GL inhibits RANKL-mediated osteoclastogenesis. Representative photographs of control (**a**,**e**), EGCG-GL-treated (**b**,**f**), RANKL-treated (**c**,**g**), and RANKL + EGCG-GL-treated (**d**,**h**) are shown. Middle magnification (**a**–**d**) and high magnification (**e**–**h**) data are shown. Scale bar: 100 µm. (**i**) Mean number of TRAP-positive multi-nucleated cells. (n = 3) *: *p* < 0.05 between samples.

**Figure 6 polymers-10-01384-f006:**
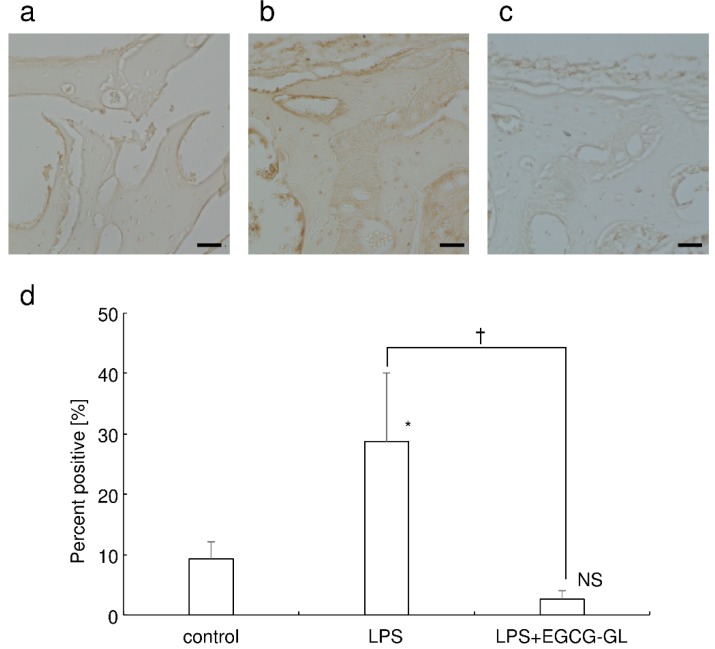
EGCG-GL attenuates LPS-mediated oxidative stress in mouse calvaria. Representative photographs of the control (**a**), LPS-treated (**b**), and LPS + EGCG-GL group (**c**) are shown. Scale bar: 100 µm. (**d**) Percent of positive staining in field. Mean values of five fields from different sections are shown. *: *p* <0.05 versus control. †: *p* < 0.05 between the samples. NS: not significant difference versus control.

**Figure 7 polymers-10-01384-f007:**
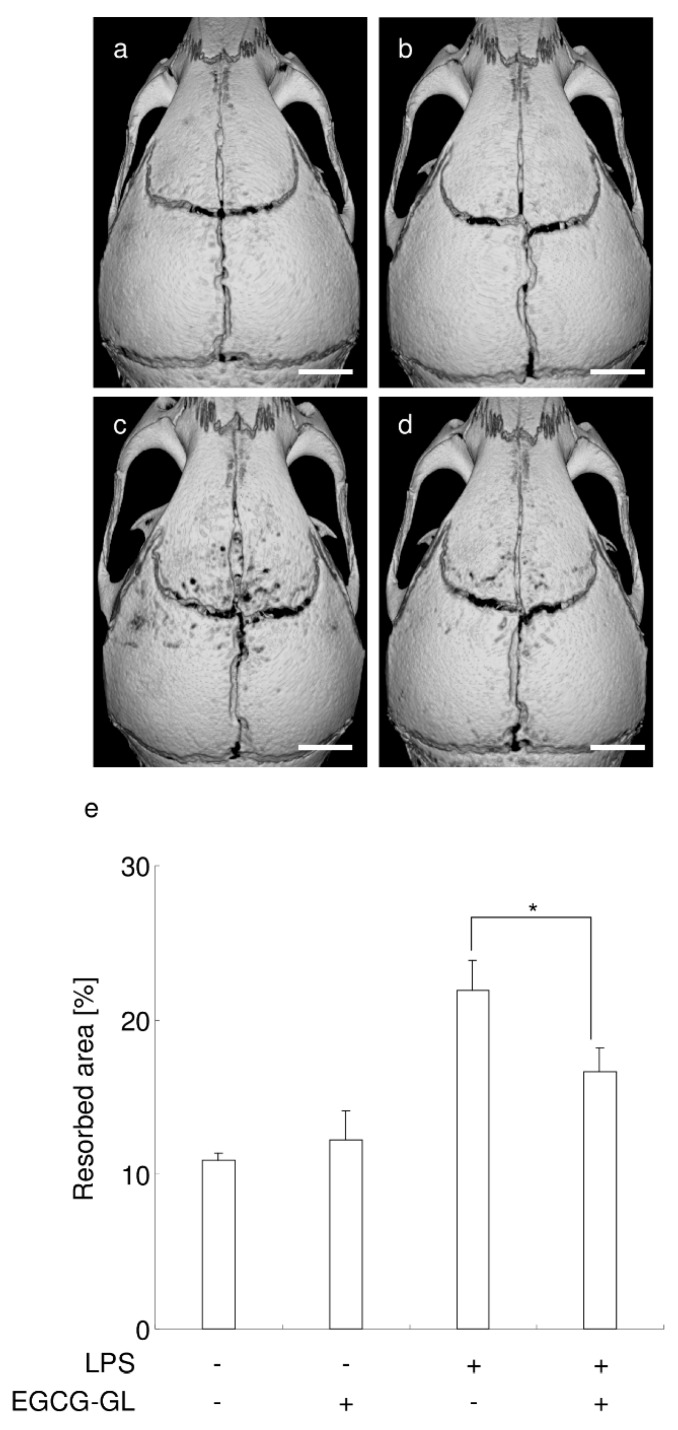
A local, single EGCG-GL injection ameliorates LPS-induced bone destruction in mice. (**a**–**d**) Representative microCT images of control (**a**), EGCG-GL-treated (**b**), LPS-treated (**c**), and LPS + EGCG-GL-treated mice (**d**). Scale bar: 1 mm. (**e**) Percentage of resorbed area in the cranial bone. *: *p* < 0.05 between groups. (n = 5 for each group).

**Figure 8 polymers-10-01384-f008:**
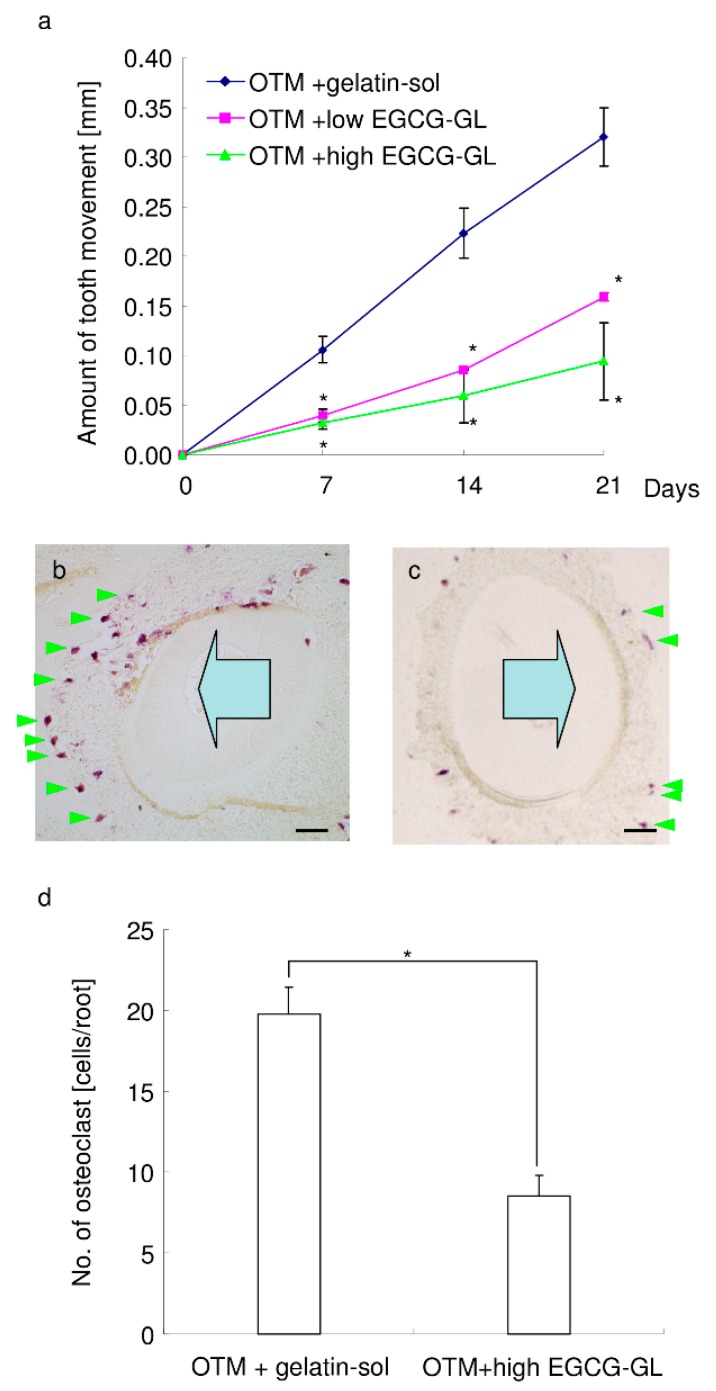
A local, single EGCG-GL injection inhibited the rate of orthodontic tooth movement (OTM) via attenuation of osteoclastogenesis. (**a**) The amount of tooth movement of the OTM + gelatin solution group (blue), OTM + low EGCG-GL group (pink), and OTM + high EGCG-GL group (green). *: *p* < 0.05 versus OTM + gelatin solution group. (n = 4, 4, and 5 for OTM + gelatin solution group, OTM + low EGCG-GL group, and OTM + high EGCG-GL group, respectively.) (**b**,**c**) Representative photographs of TRAP staining in the OTM + gelatin solution (**b**) and OTM + high EGCG-GL (**c**) groups. Arrowhead indicates osteoclasts on the alveolar bone surface. The large arrow indicates the direction of orthodontic tooth movement. Bar: 100 µm. (n = 4 and 5 for OTM + gelatin solution and OTM + high EGCG-GL, respectively.) (**d**) The number of osteoclasts on the alveolar bone surface at the compression side of mesial root of M1. *: *p* < 0.05 between groups.

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
