# Peer review of "Single Local Injection of Epigallocatechin Gallate-Modified Gelatin Attenuates Bone Resorption and Orthodontic Tooth Movement in Mice"

_polymers, 2018, doi:10.3390/polym10121384_

Round 1
Reviewer 1 Report
This revised manuscript has been improved in the quality of the content after the revision. The authors added some more experiments to comprehensively explain the findings. There is a concern on the number of osteoclasts in the positive control (Figure 5). The author detected less than 1 cell per square millimeter in positive control group and less than 0.2 cells per square millimeter in experimental group. With the nature of RAW cells in the presence of RANKL, the numbers of differentiated osteoclasts should be more than the reported finding. Would it be possible to culture the RAW cells longer to demonstrate higher number of differentiated osteoclasts and the result will be more convincing. What is the reason that the author culture the cells only 4 days instead of 7-9 days?
Author Response
Thank you very much for further fruitful suggestions to our revised manuscript. Detailed responses to the reviewers’ comments are written below.
Reviewer #1
1) There is a concern on the number of osteoclasts in the positive control (Figure 5). The author detected less than 1 cell per square millimeter in positive control group and less than 0.2 cells per square millimeter in experimental group. With the nature of RAW cells in the presence of RANKL, the numbers of differentiated osteoclasts should be more than the reported finding. Would it be possible to culture the RAW cells longer to demonstrate higher number of differentiated osteoclasts and the result will be more convincing. What is the reason that the author culture the cells only 4 days instead of 7-9 days?
---Response: Thank you very much for suggesting about the culture period for osteoclast differentiation assay. We usually performed 4 days, and it was sufficient to compare the number of TRAP positive multi-nucleated cells differentiated from RAW 264.7 cells among the different culture conditions. As the reviewer suggested, we cultured RAW cells for 7-9 days for resorption assay. In addition, osteoclast differentiation assay using mouse primary bone marrow macrophages generally takes longer culture period, such as 8 days, as compared to that of RAW 264.7 cells. Taken together, we presumed that our 4-days culture is sufficient period for osteoclast differentiation assay using RAW 264.7 cells.
Reviewer 2 Report
In “Single Local Injection of epigallocatechin gallate - modified Gelatin Attenuates Bone Resorption and Orthodontic Tooth Movement in Mice”, Katsumata et al. prepared a conjugate of epigallocatechin gallate and gelatin, and they examined its activity in vitro and in vivo. While the reporting is improved, some additional characterization and details are warranted:
How is the EGCG coupled to the gelatin? Can this be characterized chemically? What is the efficiency of this reaction?
From the starting concentration in the release study, there is 70 ug/ml EGCG, yet the ELISA is only measuring around 10 ug/ml. What is happening to the remaining EGCG?
For each figure, the number of replicates analyzed in the experiment should be indicated in the caption. For Figure 6, are the 5 fields from different replicates?
Author Response
Thank you very much for further fruitful suggestions to our revised manuscript. Detailed responses to the reviewers’ comments are written below.
Reviewer #2
1. How is the EGCG coupled to the gelatin? Can this be characterized chemically? What is the efficiency of this reaction?
---Response: Thank you very much for asking the point. In our reaction system, EGCG and gelatin were coupled with ester bond. The details of the reaction were described in our previous paper (Int. J. Mol. Sci. 2015, 16, 14143-14157; doi:10.3390/ijms160614143). Regarding the efficiency of the reaction, we tried to measure the amount of the bonded EGCG using infrared spectra, but difficult to clearly measure due to the following two reasons; 1) the peak infrared spectra of EGCG and amide linkage were so close to separate, 2) the amount of EGCG were quite few as compared to that of the gelatin. We presumed that the efficiency of the reaction would be around 70 %, judged by our data of the UV absorption value.
2. From the starting concentration in the release study, there is 70 ug/ml EGCG, yet the ELISA is only measuring around 10 ug/ml. What is happening to the remaining EGCG?
---Response: Thank you very much for confirming about our methodology. Though there are some difference in the concentration between the prepared concentration and measured concentration in EGCG solution sample, we wanted to show the difference of the concentration drop among the fractions. EGCG solution gave quick drop of the concentration from 6.6 to 0.1, and first separation (fractions 1 and 2, respectively). Fractions 3 and 4 exhibited below the sensitivity. On the other hand, EGCG-GL exhibited almost constant concentration among the fractions, 1.3, 0.7, 0.6, and 0.4 microg/mL, though the value were quite low as compared to that of the first fraction of EGCG solution. From the point of exhibiting the difference of the concentration among the fractions, we think that our methodology is accurate enough to show the difference of the concentration among the fractions.
As we previously described in the response letter, the concentration of the EGCG solution and EGCG-GL were quite different (EGCG solution: 70 microg/mL at starting concentration, around 1 microg/mL at first fraction). In this situation, we considered that the sensitivity at low concentration range is more important than the linearity within whole concentration range, and we set up the ELISA condition that exhibit quite high sensitivity at low concentration but have kind of plateau at higher concentration range. These characteristics of the ELISA would give the difference of the concentration of EGCG solution. Another possible factor for the difference between the real concentration and measured concentration at first fraction of EGCG solution would due to the non-specific binding of EGCG to the plastic ware, especially to the filtration column.
3. For each figure, the number of replicates analyzed in the experiment should be indicated in the caption. For Figure 6, are the 5 fields from different replicates?
---Response: Thank you very much for kind suggestion. We added the number of replicates in each figure legends. Yes, we measured using five fields from different sections, and this description was added in the figure legend of the figure 6.
Round 2
Reviewer 2 Report
The authors have further improved their manuscript. It is now more clear with the number of replicates being reported. I still think the release study is a weak point of the study, but I recommend publication of the manuscript.